# HAT- and HDAC-Targeted Protein Acetylation in the Occurrence and Treatment of Epilepsy

**DOI:** 10.3390/biomedicines11010088

**Published:** 2022-12-29

**Authors:** Jie Wang, Feng Yun, Jiahui Sui, Wenpeng Liang, Dingding Shen, Qi Zhang

**Affiliations:** 1Key Laboratory of Neuroregeneration of Jiangsu and Ministry of Education, Co-Innovation Center of Neuroregeneration, NMPA Key Laboratory for Research and Evaluation of Tissue Engineering Technology Products, Jiangsu Clinical Medicine Center of Tissue Engineering and Nerve Injury Repair, Nantong University, Nantong 226001, China; 2Department of Neurology & Collaborative Innovation Center for Brain Science, Ruijin Hospital Affiliated to Shanghai Jiaotong University School of Medicine, Shanghai 310000, China; 3Jiangsu Clinical Medicine Center of Tissue Engineering and Nerve Injury Repair, Research Center of Clinical Medicine, Affiliated Hospital of Nantong University, Nantong 226001, China

**Keywords:** epilepsy, acetylation, deacetylation, histone acetyltransferases (HATs), histone deacetylases (HDACs), histone deacetylase inhibitors (HDACi)

## Abstract

Epilepsy is a common and severe chronic neurological disorder. Recently, post-translational modification (PTM) mechanisms, especially protein acetylation modifications, have been widely studied in various epilepsy models or patients. Acetylation is regulated by two classes of enzymes, histone acetyltransferases (HATs) and histone deacetylases (HDACs). HATs catalyze the transfer of the acetyl group to a lysine residue, while HDACs catalyze acetyl group removal. The expression of many genes related to epilepsy is regulated by histone acetylation and deacetylation. Moreover, the acetylation modification of some non-histone substrates is also associated with epilepsy. Various molecules have been developed as HDAC inhibitors (HDACi), which have become potential antiepileptic drugs for epilepsy treatment. In this review, we summarize the changes in acetylation modification in epileptogenesis and the applications of HDACi in the treatment of epilepsy as well as the mechanisms involved. As most of the published research has focused on the differential expression of proteins that are known to be acetylated and the knowledge of whole acetylome changes in epilepsy is still minimal, a further understanding of acetylation regulation will help us explore the pathological mechanism of epilepsy and provide novel ideas for treating epilepsy.

## 1. Introduction

Epilepsy is one of the world’s most common and severe chronic neurological diseases. It is manifested as spontaneous and unprovoked recurrent abnormal discharges in the brain that might cause related complications, including neurological, psychological, cognitive, and language issues [1,2]. Any structural lesion in the brain can cause epilepsy due to its complex pathogenesis [1]. The pathological mechanism of epilepsy is an imbalance in the excitation and inhibition in the central nervous system, which is closely related to ion channel dysfunction, abnormal neurotransmitter signal transduction, synaptic structural changes, glial cell proliferation, and inflammation.

The development of the genome revolution, such as second-generation sequencing technology, has identified genes related to the occurrence of epilepsy. Different mutations in the same gene may cause varied epileptic phenotypes. The correlation between genotype and phenotype is poor, indicating the presence of other factors that affect the stability of these genes and proteins and determine the type of epilepsy that occurs. Among these, the post-translational modification (PTM) mechanism plays a vital role [1]. PTM is involved in the pathophysiological process of epilepsy, which is crucial for regulating neurobiological processes such as neural network function, synaptic plasticity, and synaptogenesis [3]. During brain development, PTMs such as acetylation, methylation, phosphorylation, ubiquitination, sulfonation, and ADP ribosylation regulate various processes, including gene regulation, cell signaling transduction, and cytoskeleton stability. Among these, acetylation is one of the most important modification methods [4,5,6].

Acetylation modification is involved in many neurobiological processes, such as neuronal death, glial cell proliferation, neuroinflammation, changes in ion channels and neurotransmitter receptors, the plasticity of axons and dendrites, and the remodeling of neural networks in the occurrence and development of epilepsy [2,3]. Increasing histone acetylation is reported to ameliorate memory and rescue cognitive impairments in neurological diseases [7]. Protein acetylation is regulated by histone acetyltransferases (HATs) catalyzing the transfer of the acetyl group to a lysine residue and histone deacetylases (HDACs) catalyzing acetyl group removal [8]. During the occurrence and development of epilepsy, the expressions of genes involved in epileptogenesis, such as immediate early genes (IEGs), are regulated by histone acetylation and deacetylation [9]. In addition to the modulation of the dynamic balance of histone acetylation and deacetylation, HATs and HDACs act on many non-histone substrates and participate in a series of neurological diseases [2]. Due to the identification of non-histone acetylation, HATs and HDACs are also referred to as lysine acetyltransferases (KATs) and lysine deacetylases (KDACs). This review primarily focuses on the changes in protein acetylation modification controlled by HATs and HDACs in the occurrence of epilepsy. Moreover, investigations of histone deacetylase inhibitors (HDACi) in the treatment of epilepsy are also summarized to provide new prospects for the clinical treatment of epilepsy. 

## 2. Protein Acetylation and Deacetylation Modifications

Acetylation is an essential form of PTM of intracellular proteins that was first discovered on lysine residues of histones [10]. To date, over 40 different lysine residues of histones have been found to be modified by acetylation [11]. With the development of proteomics, protein acetylation was discovered to be as widespread as phosphorylation, and thousands of acetylation sites have been identified [12]. Acetylation modification affects protein function through various mechanisms, such as regulating protein stability, enzyme activity, subcellular localization, and protein–protein and protein–DNA interactions. HATs, referred to as the histone “writers”, and HDACs, referred to as the histone “erasers”, are the two classes of enzymes that determine the acetylation levels, and their balance is the key regulatory mechanism of gene expression that controls different physiological processes and disease states [13]. 

### 2.1. Classification of HATs

HATs are a group of enzymes that transfer the acetyl group of acetyl coenzyme A (CoA) to the lysine residue at the N-terminal of histone. Subsequently, the negatively charged acetyl group disrupts the balance of the original histone, resulting in the dissociation of local DNA from the histone octamer and relaxing the nucleosome structure such that various transcription factors (TFs) and synergistic TFs can specifically bind to DNA binding sites and activate gene transcription [14]. Owing to the similarity in several homologous regions and acetylation-related motifs, HATs can be divided into five categories (Table 1): (1) The GCN5-related N-acetyltransferase (GNAT) family is the most classic HAT family, including GCN5 (KAT2A), p300/CBP-related factors (PCAF, KAT2B), histone acetyltransferase 1 (HAT1), elongator acetyltransferase complex subunit 3 (ELP3), and histone acetyltransferases HPA2 and HPA3 [15]. (2) The MYST family includes 60 kDa TAT-interacting protein (Tip60 and KAT5), monocytic leukemia zinc finger protein (MOZ and KAT6A), MOZ-related factors (MORF and KAT6B), histone acetyltransferase (HBO1 and KAT7) binding to initiation recognition complex 1 (ORC1), an ortholog of Drosophila males-absent on the first (MOF and KAT8), and histone acetyltransferases (SAS2, SAS3, and ESA1). The classification of this family is based on the presence of a highly conserved MYST domain, which consists of an acetyl CoA-binding motif and a zinc finger [16,17]. (3) The P300/cAMP response element binding protein (CBP or CREB) family (KAT3A and KAT3B), which is closely related to cell differentiation and apoptosis, has multiple non-histone substrates. Members of this family have four separate transactivation domains, including cysteine–histidine-rich region 1, kinase-induced domain interacting with CREB, another cysteine–histidine-rich region, and a nuclear receptor coactivator binding domain. P300/CBP is a coactivator of several TFs that connect chromatin remodeling and transcription processes to coordinate physiological activities in vivo, such as signal transduction [18,19]. (4) The transcription initiation factor TFIID 230/250 kDa subunit (TAFII230/250) family is an integral part of the TF complex TAFIID in humans [20]. (5) Other HATs, including α-tubulin N-acetyltransferase 1 (ATAT1), catalyze α-tubulin acetylation and promote axonal transport [21]. Various HATs play different roles in cells, and their stable expression is crucial to maintain the physiological activities of the cells.

### 2.2. Classification of HDACs

HDACs are a class of proteases that remove acetyl groups from lysine residues, subsequently restoring their original positive charges. Then, the HDACs are tightly bound to the negatively charged DNA, making it difficult for TFs to access the promoter, thereby inhibiting the transcription of specific genes [3]. HDACs are divided into four classes (Table 1): Class I (1, 2, 3, and 8), class IIa (4, 5, 7, and 9), and class IIb (6 and 10) are designated according to their functional and structural similarities to yeast deacetylase proteins, while class III constitutes a family of sirtuin proteins, including SIRT1-7, and class IV includes HDAC11 [54,107]. Class I, II, and IV HDACs are a family of zinc-ion-dependent enzymes, while class III HDACs are a family of nicotinamide adenine dinucleotide (NAD^+^)-dependent deacetylases [108]. These HDACs have different distribution patterns in the brain, which are intersecting and specific [109,110,111]. Moreover, several differences have been detected in the distribution of HDACs in various types of cells, most of which are expressed in neurons, and the proportion in the nucleus is greater compared to the cytoplasm and neurites [112]. In addition to the expression of HDAC3 in the nucleus and cytoplasm, class I HDACs are expressed in the nucleus and have high enzyme activity. Class IIa HDACs can shuttle between the nucleus and cytoplasm in response to various cell signal responses, with cell and tissue specificity, while class IIb HDACs mainly exist in the cytoplasm [113]. Class I and II HDACs are widely expressed in the whole brain and are closely related to the occurrence of epilepsy [114]. Class III HDACs are located in the nucleus, cytoplasm, and mitochondria and play a critical role in metabolic regulation [54,115]. The class IV HDAC is expressed in the brain, heart, muscle, kidney, and testis; however, its function is poorly understood [108,116]. In addition to using histones as the substrate for deacetylation, HDACs can also reduce non-histone acetylation levels and regulate various neurobiological processes by catalyzing some non-histones, such as TFs, cytoskeletal proteins, and molecular chaperones (Hsp90 and Hsp70) [2].

## 3. Substrates of HATs and HDACs

### 3.1. Histone Substrates

Histones are rich in positively charged basic amino acids and can interact with negatively charged phosphate groups in the DNA to form DNA–histone complexes. According to the amino acid composition and molecular weight, histones are divided into five categories: H1, H2A, H2B, H3, and H4. The dimers of four histones, H2A, H2B, H3, and H4, form the core of nucleosomes. H1 binds to DNA wounds between nucleosomes [117,118,119] (Figure 1). The acetylation and deacetylation of lysine residues of histones H2A, H2B, H3, and H4 are regulated by various HATs and HDACs, as shown in Table 1.

### 3.2. Non-Histone Substrates

The results of a proteomic analysis showed that the high frequency of non-histone acetylation constitutes the main component of acetylated proteins in mammalian cells [120]. Typically, non-histone acetylation involves key physiological and disease-related cellular processes, such as gene transcription, DNA damage repair, cell division, signal transduction, protein folding, autophagy, and metabolism [120]. Acetylation involves the regulation of more than 100 kinds of non-histone proteins, including TFs, transcriptional coactivators, nuclear receptors, molecular chaperones, and signal transduction molecules, which are also regulated by various HATs and HDACs listed in Table 1 (Figure 1).

## 4. Altered Protein Acetylation in Epileptogenesis 

Multiple methods have been developed to determine protein acetylation levels [121]. Western blotting analyses or radiolabeling methods are usually used for the semi-quantification of acetylated proteins. However, these methods only detect the proteins known to be acetylated and peptides enriched by immunoprecipitation. As the protein acetylation networks regulated by HATs and HDACs are complicated, the understanding of the characterization of the brain acetylome will be helpful to investigate the underlying cellular and molecular mechanisms involved in neurological diseases. With the advances in mass spectrometry (MS)-based quantitative proteomics, the relative or absolute quantification of acetylated proteins has been applied to understand physiological and pathological conditions [121]. The mouse and rat brain acetylomes have been profiled, which will contribute to understanding the molecular mechanisms of protein acetylation related to the development and disorders of the central nervous system [122,123]. For example, a cohort-scale histone-acetylome-wide association study (HAWAS) was performed to study the histone acetylome in autism spectrum disorder (ASD), providing a foundation for studies of histone modification changes in other neurological diseases [124]. So far, our knowledge of the brain acetylome in epilepsy is still minimal. In a kainic acid (KA) model of temporal lobe epilepsy (TLE), altered mitochondrial acetylation profiles were investigated with MS, and 68 acetylated proteins were only observed in the KA model [125]. However, most of the research on acetylation modifications in epilepsy has focused on the differential expression of H3 or H4 acetylation. On the other hand, the expressions of two classes of enzymes, HDACs or HATs, are usually detected to indirectly reflect the acetylation modifications. 

### 4.1. Alteration and Regulation of Histone Acetylation in Epileptogenesis

Histone acetylation is one of the epigenetic mechanisms involved in gene expression and has been linked to epileptic disorders. The expression of many IEGs, such as *c-fos* and *c-jun*, is increased in epilepsy models due to enhanced histone acetylation, which is regulated by the balance between HDACs and HATs [3,126,127]. Several studies have reported that the changes in the expression of class I, II, and IV HDACs in the two mouse TLE models are significantly different during chronic epilepsy and after acute epileptic seizures (SE) [128,129]. During KA-induced acute SE, the mRNA expression of *HDAC1*, *HDAC2*, and *HDAC11* in the hippocampal granule cells and the pyramidal cell layer is decreased significantly. Similar changes have been detected after pilocarpine-induced SE, while decreased HDAC3 and HDAC8 expression was observed during chronic epilepsy [128]. However, the protein expression patterns of HDACs are missing due to the unspecific antibodies. In addition, the relationship between the expression of HDACs and acetylation modifications remains to be corroborated. In the two epilepsy models above, the increases in *c-fos*, *c-jun,* and brain-derived neurotrophic factor *(BDNF)* expression were attributed to the decreased expression of class I HDACs, the increased phosphorylation of histone H3, and the excessive acetylation of histone H4 in rat hippocampal neurons [126,130,131]. 

On the other hand, the increase in HAT expression and activity is also related to the IEG expression in epilepsy models [132]. CBP is a well-known transcriptional coactivator with intrinsic HAT activity. Pretreatment with curcumin, an HAT inhibitor specific for CBP/p300, decreases H4 acetylation of the *c-fos* gene and reduces IEG expression and the severity of KA-induced SE. Conversely, HDACi led to excessive acetylation of histones and increased IEG expression after KA administration [127,131]. As many other cellular effects of curcumin have been reported, other pathways leading to the suppression of histone modifications could not be ruled out. Collectively, the above research suggests that histone acetylation modification plays a key role in IEG expression and the development of epilepsy.

Some other genes associated with epileptogenesis were also reported to be regulated by histone acetylation, such as cAMP response element binding protein (*CREB*), glutamate decarboxylase (*GAD*), and N-methyl-d-aspartate (NMDA)-receptor-related genes. *CREB* is a major transcriptional activator that regulates a wide range of cellular processes through calcium signaling, which led to the differential expression of γ-aminobutyric acid type A receptor (GABA_A_R) subunits in the hippocampus in an epilepsy model [133]. The *CREB* gene promoter is selectively modified by H3 and H4 in the electric-convulsion-induced epilepsy model. In this model, a decrease in the H4 acetylation level leads to the chronic downregulation of *c-fos* transcription, while an increase in H3 acetylation contributes to the chronic upregulation of *BDNF* transcription [126]. As the most important inhibitory neurotransmitter closely related to epilepsy, γ-aminobutyric acid (GABA) synthesis is catalyzed by GAD, and the production process is also regulated by acetylation modification [134,135]. In TLE patients and pilocarpine-induced epilepsy models, decreased expression of *GAD* is associated with decreased H3 acetylation in hippocampal neurons. Treatment with HDACi significantly reversed the reduction in H3 acetylation on the *GAD* promoter and restored the downregulation of GAD at both the protein and mRNA levels [135]. Repeated electroconvulsive seizures upregulate HDAC2 expression in the rat frontal cortex and downregulate HDAC2-targeted NMDA-receptor-related genes through the acetylation of H3 and/or H4. A ChIP assay indicated a significant increase in the *HDAC2* occupancy in the promoters of these downregulated genes [136].

Taken together, these results suggest that histone acetylation may be involved in epileptogenesis. However, all above mentioned data related to acetylation detection were achieved by Western blotting, immunofluorescence staining, and ChIP assays, which could only identify individual histone acetylation sites or proteins in the experiments. In addition, most studies detected acetylation modifications after seizure onset, which might be the outcomes of the occurrence of seizures rather than the primary changes that cause epilepsy. Further research should pay more attention to the changes before the first epileptic seizure and during the time course of the development. 

### 4.2. Regulatory Role of Non-Histone Acetylation in Epileptogenesis

#### 4.2.1. TFs

In neurons, *p53* is a pleiotropic transcription factor that controls DNA repair, cell cycle progression, differentiation, and apoptosis, and its over-expression induces death in hippocampal neurons [137,138]. The level of *p53* in the hippocampal neurons of patients with epilepsy is elevated, while inhibiting *p53* activation in animal models can prevent neuronal cell death [139,140]. The transcriptional activation of *p53* is regulated by several HDACs. HDAC1 acetylates p53 and inhibits the transcriptional activation of the *p53* gene both in vitro and in vivo, thus reducing neuronal apoptosis after seizures [141]. In the nucleus, immunofluorescence and ChIP assays indicated *p53* as a critical non-histone substrate of HDAC4, and its role is related to neuronal death after epilepsy [142]. SIRT1, 2, and 3 can deacetylate p53 in the nucleus, cytoplasm, and mitochondria, respectively, thus promoting cell survival and inhibiting aging and apoptosis [143]. In a rat lithium-pilocarpine epilepsy model, SIRT1 was reported to regulate seizures and neuron death during epileptogenesis via the deacetylation of p53 [144]. In another epilepsy model, targeting p53 may have a negative effect. It was reported that the inhibition of p53 by HDAC4 after DNA damage mediated the deacetylation of the p53 C-terminal lysine residue, detected by Western blotting and immunofluorescence staining, which might lead to a severe epileptic phenotype. However, the influence of the genetic background could not be entirely ruled out in this study [145].

Myocyte enhancer factor 2 (MEF2) is one of the nuclear substrates of HDAC3, which was confirmed by co-immunoprecipitation (Co-IP), immunofluorescence, and ChIP assay [146]. HDAC3 interacts with and deacetylates the MADS-box domain of MEF2, thereby inhibiting MEF2-dependent transcription. Several studies have reported the role of MEF2 and its target genes in epilepsy. In both TLE patients and rat models, MEF2 expression is significantly downregulated in the temporal neocortex, suggesting its involvement in the pathogenesis of TLE [33,147]. 

#### 4.2.2. Signaling Pathway Molecules

The TGF-β/SMAD signaling pathway has been implicated in the pathophysiology of epilepsy. SMAD proteins play a key role in the transduction of TGF-β signaling from cell surface receptors to the nucleus [148,149]. Serine–threonine kinase receptor associated protein (STRAP) and SMAD7 act synergistically in inhibiting TGF-β signaling as negative regulators and antagonists, respectively [150]. The acetyltransferase p300 protects SMAD7 from TGF-β-induced degradation by acetylating two lysine residues of SMAD7, detected by immunoprecipitation and immunoblotting, while HDAC1 mediates the deacetylation of SMAD7 and thus decreases its stability [150,151]. In the pilocarpine-induced SE model, the protein levels of STRAP and SMAD7 are significantly reduced in the rat hippocampus and temporal cortex, indicating that the acetylation modification of SMAD might be involved in epileptogenesis [150]. 

High-mobility group box-1 (HMGB1) and Toll-like receptor 4 (TLR4) are increased in human epileptogenic tissue and a mouse model of chronic seizures, indicating that the HMGB1/TLR4 axis is involved in generating and perpetuating seizures [152,153]. Lipopolysaccharide (LPS) treatment can promote HDAC4 degradation, leading to the acetylation of HMGB1, which prevents HMGB1 entry into the nucleus and triggers inflammation [154]. As one of the downstream molecules of HMGB1/TLR4, NF-κB, composed of p50 and p65 subunits, is a cytoplasmic target of HDAC3. A combination of approaches had identified NF-κB pathway activation during SE, and one mechanism for the effect of NF-κB activation is through the regulation of BDNF expression [155]. HDAC3 can deacetylate p65 to regulate NF-κB activity [156]. In addition, SIRT2 can also interact with p65 in the cytoplasm and deacetylate p65 at lys310, which in turn regulates the expression of NF-κB-dependent genes [157]. It has been reported that the activation of the HMGB1/TLR4/NF-κB pathway is closely associated with the acetylation of HMGB1 in the hippocampus following cold exposure [158]. Whether this is a common phenomenon during other stress conditions in the central nervous system, such as epilepsy, needs further investigation.

#### 4.2.3. Chaperones

The critical role of Hsp70 in regulating neuronal excitability makes it a widely used indicator of stress in the acute phase of epilepsy [159]. Hsp70 is upregulated in a KA model of TLE, leading to a proteasomal degradation that mediates related signaling pathways involved in epilepsy, while the inhibition of Hsp70 suppresses neuronal hyperexcitability and attenuates epilepsy [24,160]. The acetylation of Hsp70 in the cytoplasm was regulated by HDAC10 in human neuroblastoma cell lines, which was confirmed by a Co-IP experiment [161]. In another study, Hsp70 was co-immunoprecipitated with class I HDACs, indicating that the acetylation modification of Hsp70 might also be associated with class I HDACs [160]. Both HDACi and HDAC6 knockdowns induced Hsp90 acetylation at the K294 site and reduced its chaperone activity [48]. Studies on animal models and human samples have shown that Hsp90 expression is elevated in epilepsy, and the inhibition of Hsp90 is also considered an effective treatment for epilepsy [48,162]. Further studies based on MS should be performed to identify more acetylation sites in these chaperones to regulate their functions by acetylation and deacetylation modifications.

## 5. Role of HDACi in the Treatment of Epilepsy 

### 5.1. Classification of HDACi

HDACi are a small molecules with an HDAC-inhibitory activity that modulates biological functions by rebuilding or increasing the acetylation levels of lysine residues in nuclear and cytoplasmic proteins [163,164]. HDACi stimulate gene transcriptional activity in cells by regulating the level of histone acetylation. On the other hand, it also promotes the acetylation of non-histone proteins and regulates the interactions, intracellular localization, and stability of these proteins [163,165]. HDACi are divided into four major classes according to the chemical structures (Table 2): hydroxamates, benzamide derivatives, cyclic peptides, and short-chain fatty acids (SCFAs) [166].

Hydroxamates, with short half-lives but long-lasting effects, have the largest molecular weights among the four types of HDACi. As a class of broad-spectrum HDACi, hydroxamates exhibit low selectivity for the inhibition of HDACs, including Trichostatin A (TSA), Vorinostat (SAHA), Panobinostat (LBH589), Belinostat (PXD101), Givinostat (ITF2375), Abexinostat, and Dacinostat [166]. Class I and II HDACs can be inhibited by these compounds. SAHA and TSA are the two most commonly used HDACi hydroxamates in clinical practice [167]. Benzamide derivatives are a class of HDACi with long half-lives, including Entinostat and Mocetinostat. Cyclic peptides include Romidepsin (FK228), Apicidin, and Cycloheximide. SCFAs include small, structurally simple compounds, such as valproic acid (VPA), sodium butyrate (NaB), and sodium phenylbutyrate (4-PBA), among which VPA is a commonly used antiepileptic drug in the clinic.

Different HDACi have specific effects on various classes of HDACs. Currently, as a potential drug or preparation, alone or in combination with other preparations, HDACi have been widely used in animal models of disease and clinical treatment in humans. Some HDACi are expected to become good antiepileptic drugs [168,169] (Figure 2).

### 5.2. Role of HDACi in Epilepsy Treatment

As an antiepileptic drug, VPA inhibits epilepsy by increasing the level of GABA in the brain. Typically, it is a critical broad-spectrum HDACi [168]. Therefore, it combines the characteristics of anticonvulsant and epigenetic regulatory drugs. VPA treatment can significantly protect cholinergic and GABAergic neurons from excitotoxic injury. The long-term use of VPA increases the histone H3 acetylation level in the brain [170]. The granule cells produced by seizures may interfere with hippocampal function and lead to cognitive damage caused by epileptic activity in the hippocampal circuit. VPA can block epilepsy-induced neurogenesis, which might be mediated by inhibiting HDACs and normalizing HDAC-dependent gene expression in the hippocampal dentate gyrus [171].

Tuberous sclerosis (TSC) is a genetic disease characterized by seizures, autism, and some cognitive deficits [172]. Studies on TSC2 mutant heterozygous mice (TSC2^+/−^) showed that hippocampal histone H3 acetylation was reduced and had abnormal synaptic plasticity and seizure tendency, while attenuating HDAC activity restored normal neurological function [172]. HDACi, such as VPA, TSA, and SAHA, could restore the histone H3 acetylation level, improve synaptic plasticity, and reduce the epileptic phenotype in a TSC2^+/−^ mice model [172]. The limitations of this study were that the mechanism involved in the global reduction in histone acetylation levels has not been explored and the potential for diminished HAT activity cannot be eliminated. 

NaB is an HDACi of class I/IIa that is used in the treatment of epilepsy. In a rat hippocampal kindling epilepsy model, daily NaB treatment significantly inhibited HDAC activity and prevented the development of epileptogenesis [173]. NaB was reported to increase the acetylation state of H3 and H4 histones in the mouse hippocampus and cerebral cortex, accompanied with an increase in IEGs [174]. In the WAG/Rij epilepsy rat model (genetic model of absence epilepsy with mild depression), the decreased acetylation level of mesencephalic histones H3 and H4 was significantly increased after the administration of NaB, VPA, and their combination, with decreased expression of HDAC1 and 3, demonstrating the antiepileptic effect of NaB and VPA [175]. In the stress state, NaB can regulate the function of MK-801, a non-competitive NMDA receptor antagonist, to delay the occurrence of epilepsy in mice and reduce the severity of seizures [176]. In addition, NaB has a long-term effect on functional improvement, even after the cessation of the treatment [173]. These findings suggest that NaB has a strong antiepileptic effect. 

TSA is a selective HDACi of class I/II. In a rat model of pilocarpine-induced SE, the acetylation of H4 in rat hippocampal CA3 neurons was reduced at the glutamate receptor 2 (*GluR2* or *GRIA2*) promoter but increased at the *BDNF* promoter P2 after the induction of SE by a ChIP assay. TSA could quickly prevent and reverse the deacetylation of GluR2-associated histones as well as blunt the seizure-induced downregulation of *GluR2* mRNA in CA3, thus reversing the seizure-induced neuronal damage [130]. In a KA-induced acute seizure model, treatment with TSA dramatically elevated Neuregulin 1 (Nrg1) expression via increasing the acetylated occupancy of H3 and H4 at the *Nrg1* promoter locus to suppress seizures [177].

SAHA is one of the broad-spectrum HDACi that inhibits class I, II, and IV HDACs. In a rat model of KA-induced seizures, SAHA pretreatment increases seizure latency and decreases the seizure score. It also inhibits the microglial activation and neuronal apoptosis caused by epilepsy. SAHA reduces the acetylation level of H3K9, regulates and inhibits the TLR4/MyD88 signal transduction induced by epilepsy, inhibits *TLR4* gene expression, and prevents brain injuries caused by epilepsy [178]. In a model of intractable zebrafish epilepsy induced by a GABA_A_R γ2 subunit (*GABRG2*) mutation, SAHA reduces the expression of HDAC1 and HDAC10 and inhibits zebrafish seizures in a dose-dependent manner. Compared to traditional AEDs, it has a short onset time and has a better curative effect in intractable epilepsy caused by a *GABRG2* mutation [179]. These observations indicate that SAHA is an effective neuroprotective and anticonvulsant drug that has a potential application in epilepsy treatment.

So far, most works used HDACi or RNA silencing to investigate the role of acetylation in epilepsy. Indeed, most HDACi that have been applied are non-specific, and some of them might display pharmacological effects other than regulating protein acetylation. Alternatively, HDAC knockout mice would be a useful tool to determine the involvement of HDACs in epilepsy. However, research in this area is still very limited. Another problem that should be noticed is that there are many types of epilepsy, and the epileptogenic processes may not be identical in all of them. Thus, the investigation of the acetylation modifications in each type of epilepsy will be very important for precise treatments.

## 6. Conclusions

This review summarizes the changes in acetylation modification in epileptogenesis and the regulatory role of HDACi in the treatment of epilepsy. Acetylation modification on both histone and non-histone substrates regulates a series of neurological functions by regulating protein stability, enzyme activity, subcellular localization, and protein–protein and protein–DNA interactions, thus playing a critical role in the occurrence of epilepsy. Currently, many HDACi have been developed to target various classes of HDACs, and some of them are expected to become effective antiepileptic drugs with neuroprotective and anticonvulsant effects. An in-depth understanding of acetylation regulation would be valuable in exploring the pathological mechanism of epilepsy and providing new ideas for the treatment of epilepsy. Additional studies of protein acetylation with quantitative proteomics methods might provide new prospects for the clinical treatment of epilepsy.

## Figures and Tables

**Figure 1 biomedicines-11-00088-f001:**
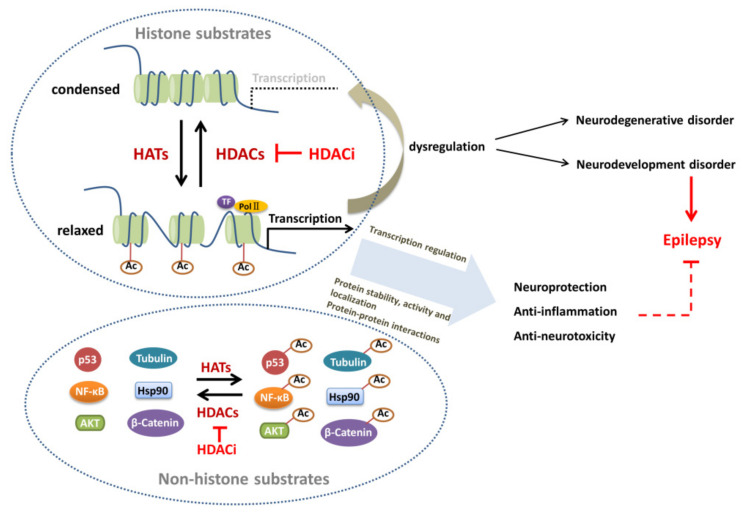
The mechanisms of protein acetylation/deacetylation modifications in neurological diseases. Transcriptional disorders are often involved in neurodegenerative diseases and neurodevelopmental disorders regulated by HATs and HDACs. Typically, DNA is wound around histone to form a nucleosome structure. When HATs transfer the acetyl group to the lysine residue of histone, the negatively charged acetyl group breaks the balance of the original histone, resulting in the dissociation of local DNA from the histone octamer and the relaxation of the nucleosome structure such that the various TFs and synergistic TFs can specifically bind to DNA binding sites and activate gene transcription by RNA polymerase II (pol II). HDACs are a class of proteases that remove acetyl groups from lysine residues. When HDACs remove the acetyl group in the histone lysine residues, the histone restores its original positive charge and binds tightly to the negatively charged DNA, making it difficult for TFs to access the promoter, thereby inhibiting the transcription of specific genes. In addition, HATs and HDACs also act on many non-histone substrates, such as TFs (p53 and MEF2), chaperones (Hsp90 and Hsp70), signaling pathway molecules (SMAD and NF-κB), and protease. The key regulatory mechanism of gene expression is the balance between the effects of HATs and HDACs. The acetylation/deacetylation modifications affect protein function through various mechanisms, including transcriptional regulation, the regulation of protein stability, enzyme activity, subcellular localization, and protein–protein interaction. The dysfunction of protein acetylation/deacetylation modifications may lead to epileptogenesis. HDACi can inhibit the activity of HDACs and promote antiepileptic functions via neuroprotective, anti-inflammatory, and anti-neurotoxic effects.

**Figure 2 biomedicines-11-00088-f002:**
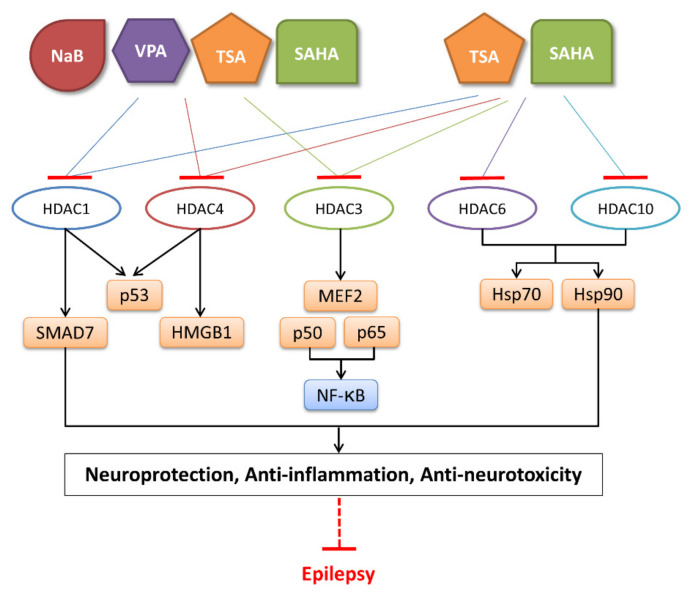
The applications of HDACi in epilepsy treatment. The HDACi that have been used in animal models of epilepsy and clinical treatment in humans and their potential targets are illustrated. NaB and VPA inhibit HDAC1, 3, and 4 in epilepsy models, while TSA and SAHA broadly target HDAC1, 3, 4, 6, and 10. SMAD7, HMGB1, and p53 are substrates of HDAC1 and HDAC4. MEF2 and NF-κB (p50 and p65) are substrates of HDAC3. Hsp70 and Hsp90 are substrates of HDAC6 and HDAC10. By regulating the acetylation of the above substrates, HDACi might promote neuroprotective, anti-inflammatory, and anti-neurotoxic effects to inhibit epileptogenesis.

**Table 1 biomedicines-11-00088-t001:** Substrates of HDACs and HATs.

	Class/Family		Members	Localization	Histone Substrates	Non-Histone Substrates
**HDACs**	Class I		HDAC1	Nucleus	H3K9ac, H3K14ac, H3K18ac, H3K23ac, H3K27ac [22]	p53 (K382ac) [23], Hsp70 [24], STAT3 [25], E2F-1 [26]
			HDAC2	Nucleus	H3K9ac, H4K12ac [27]	p53 (K320ac) [28], Nrf2 [29], Bcl-6, STAT3 [30]
			HDAC3	Nucleus Cytoplasm	H3K9ac, H3K14ac, H4K5ac, H4K12ac [31]	p65 (K310ac, K314ac, K315ac) [32], MEF2 [33], STAT1, STAT3 [34], GATA1 [35]
			HDAC8	Nucleus Cytoplasm	H3K9ac, H3K14ac, H3K56ac [36]	ARID1A (K1808ac) [37], cortactin [38]
	Class II	Class IIa	HDAC4	Nucleus		p53, STAT1 [39], SRF [40], HIF-1α [41], ATF4 [40], FoxO1 [42]
			HDAC5	Nucleus		reelin [43]
			HDAC7	Nucleus		HIF-1α [44]
			HDAC9	Nucleus		
		Class IIb	HDAC6	Cytoplasm	H3K14ac, H4K5ac, H4K12ac, H4K16ac [45]	G3BP1 (K376ac) [46], cortactin (K124ac) [47], Hsp90 (K294ac) [48], β-Catenin [49], Prx I, Prx II [50], Survivin [51], AKT (K37ac, K163ac) [51]
			HDAC10	Cytoplasm		MSH2 [52], MMP2, MMP9 [53], Hsp70 [24]
	Class III		SITR1	Nucleus	H3K9ac, H3K14ac, H3K56ac, H4K16ac [54]	BMAL1 [55], TGF-β [56], PGC-1α [57], p53 (K379ac) [58]
			SIRT2	Nucleus Cytoplasm	H3K56ac, H4K16ac [54]	G6PD [38], α-tubulin [59]
			SIRT3	Nucleus Mitochondria	H4K16ac [54], H3K4bhb, H3K9bhb, H3K18bhb, H3K23bhb, H3K27bhb, H4K16bhb [60], H3K4cr [61]	ACSS1 (K642ac) [62], SDHA (K179ac) [63]
			SIRT4	Mitochondria		MCCC [64], MCD [65]
			SIRT5	Mitochondria		PKM2 (K311ac) [66], CPS1 [67]
			SIRT6	Nucleus Endoplasmic Reticulum Mitochondria	H3K9ac, H3K56ac [68]	TNF (K19ac, K20ac) [69]
			SIRT7	Nucleus Cytoplasm	H3K18ac [70]	FKBP51 (K28ac, K155ac) [71], Ran (K37ac) [72], U3-55k (K12ac, K25ac) [73], nucleophosmin (K27ac, K54ac) [74]
	Class IV		HDAC11	Nucleus		IL-10 [75]
**HATs**	GNAT		GCN5 (KAT2A)	Nucleus	H3K9 [76]	PLK4 (K45, K46) [77], ISWI (K753) [78], SNF2 (K1493, K1497) [79], TBX5 [77]
			PCAF (KAT2B)	Nucleus	H3K9 [76]	ISX (K69) [80], IDH2 (K180) [81]
			HAT1 (KAT1)	Nucleus Cytoplasm Mitochondria	H4K5, H4K12 [45]	TPR (K531) [82]
			ELP3	Nucleus Cytoplasm		Bruchpilot [83]
			HPA2	Cytoplasm		Polyamines, small basic proteins [84]
			HPA3	Cytoplasm		Polyamines, D-amino acids [84]
	MYST		Tip60 (KAT5)	Nucleus Cytoplasm	H2AK5, H2AK15, H3K14, H4K5, H4K8, H4K12, H4K16 [76]	BMAL1 (K538) [85], ATM (K3016) [86], Ran, Pacer [87]
			MOZ (KAT6A)	Nucleus	H3K9, H3K14, H3K23 [76]	p53 (K120, K382) [88], BRPF1 [89]
			MORF (KAT6B)	Nucleus	H3K14, H3K23 [76]	BRPF1 [89], Runx2 [90]
			HBO1 (KAT7)	Nucleus	H3K9, H3K14, H4K5, H4K8, H4K12 [76]	CDT1 [91]
			MOF (KAT8)	Nucleus Mitochondria	H4K5, H4K8, H4K16 [92]	Lamin A/C (K311) [93]
			SAS2	Nucleus		
			SAS3	Nucleus		
			ESA1	Nucleus		Atg3 [94]
	p300/CBP		p300 (KAT3B)	Nucleus Cytoplasm	H2AK4, H2AK5, H2AK7, H2AK9, H2AK11, H2AK13, H2BK5, H2BK11, H2BK12, H2BK15, H2BK16, H2BK20, H2BK21, H2BK23, H2BK24, H3K18, H3K27, H3K36, H4K5 [95]	Sam68, hnRNP M [96], Snail, Smad4, PCNA [97], FoxO1 [98], GATA1 [99], NCOA1, NCOA2, NCOA3, ARNT, ARNT2 [95]
			CBP (KAT3A)	Nucleus Cytoplasm	H2AK4, H2AK5, H2AK7, H2AK9, H2AK11, H2AK13, H2BK5, H2BK11, H2BK12, H2BK15, H2BK16, H2BK20, H2BK21, H2BK23, H2BK24, H3K18, H3K27, H3K36, H4K5 [95]	TPX2 (K75, K476, and K582) [100], GATA1 [101], NCOA1, NCOA2, NCOA3, ARNT, ARNT2 [95], Snail, FoxO1, PCNA [102]
	TAFII230/250		TAFII250	Nucleus		TAF1
	Others		ATAT1	Nucleus Cytoplasm		α-Tubulin (K40) [103]
			ESCO1	Nucleus Cytoplasm		SMC3 (K105, K106) [104]
			ESCO2	Nucleus Cytoplasm		SMC3 (K105, K106) [104]
			CLOCK (KAT13D)	Nucleus Cytoplasm		ASS1 (K165, K176) [105]
			ATAC2 (KAT14)	Nucleus Cytoplasm	H3K9, H4K5, H4K12, H4K16 [106]	

**Table 2 biomedicines-11-00088-t002:** Classification of HDACi.

Classification	Compounds	Structure	IC50	HDAC Targets	Therapeutic Potential
Hydroxamic acids	Trichostatin A (TSA)	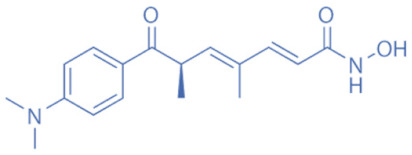	1.8 nM	Class I and II	Inhibition of breast cancer cell line proliferation
	Vorinostat (SAHA)	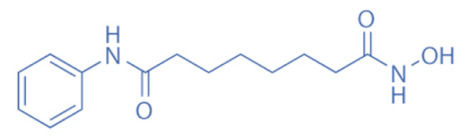	~10 nM	Class I and II	Inhibition of tumor growth
	Panobinostat (LBH589)	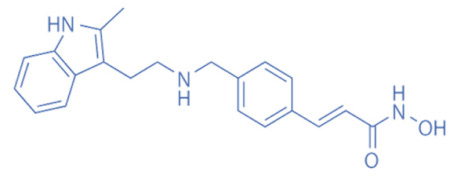	5 nM	Class I and II	Both autophagy and apoptosis can be induced, effectively disrupting the latency of HIV in vivo.
	Belinostat (PXD101)	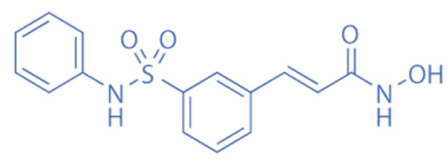	27 nM	Class I and II	Inhibition of tumor cell growth
	Givinostat (ITF2357)	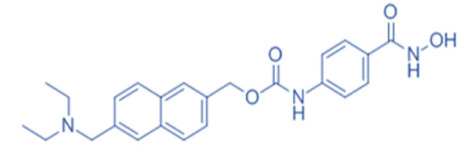	10 nM (HD2), 16 nM (HD1-A), 7.5 nM (HD1-B)	Class I and II	Improving islet cell survival and action on multiple myeloma cell lines
	Abexinostat (PCI-24781)	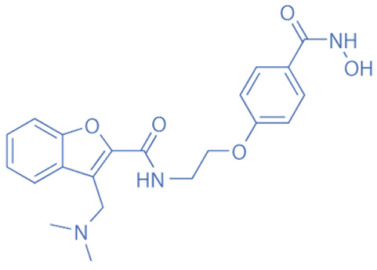		Class I and II	Anti-neoplasmic activity
	Dacinostat (LAQ824)	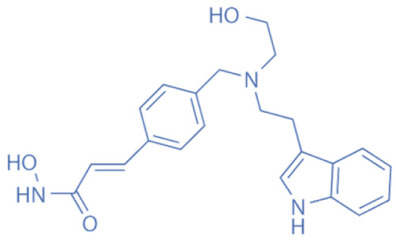	32 nM	Class I and II	Inhibition of cancer cell growth
Benzamides	Entinostat (MS-275)	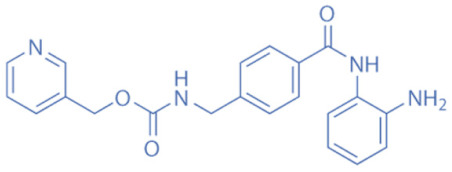	0.51 μM (HDAC1), 1.7 μM (HDAC3)	HDAC1, HDAC2, HDAC3	Induction of autophagy and apoptosis in tumor cells
	Mocetinostat (MGCD0103)	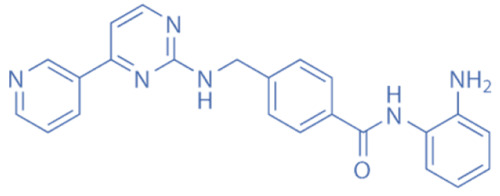	0.15 μM (HDAC1), 0.29 μM (HDAC2), 1.66 μM (HDAC3), 0.59 μM (HDAC11)	HDAC1, HDAC2, HDAC3, HDAC11	Anticancer activity
Cyclic peptides	Romidepsin (FK228, Depsipeptide)	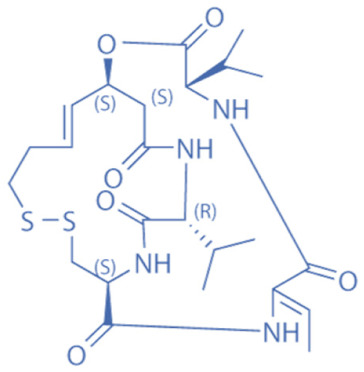	36 nM (HDAC1), 47 nM (HDAC2)	HDAC1, HDAC2	Inhibition of growth and induced apoptosis in neuroblastoma tumor cells
	Apicidin	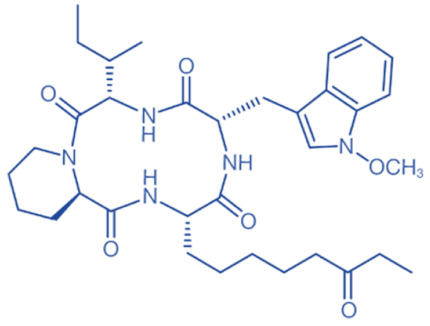	0.7 nM	HDAC1, HDAC4, HDAC8	Inhibition of tumor cell proliferation
	Cycloheximide (NSC-185)	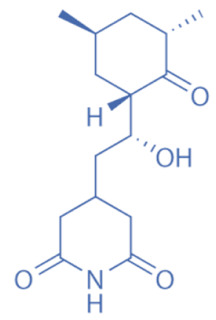	532.5 nM (protein synthesis), 2880 nM (RNA synthesis)	HDACs associated with protein synthesis	Inhibition of iron death and autophagy
SCFAs	Valproic Acid (VPA)	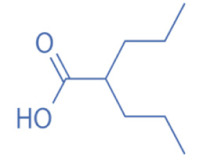	0.4 mM	Class I and IIa	Treatment of epilepsy
	Sodium Butyrate (NaB)	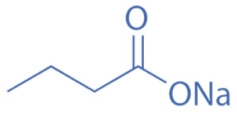		Class I and IIa	Inhibition of cell cycle progression in cancer cells, promotion of differentiation, and induction of apoptosis and autophagy
	Sodium Phenylbutyrate (4-PBA)	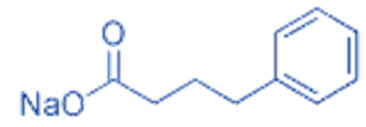		Class I and IIa	Induction of apoptosis in prostate cancer cells

## Data Availability

Not applicable.

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
