# Peer review of "HAT- and HDAC-Targeted Protein Acetylation in the Occurrence and Treatment of Epilepsy"

_biomedicines, 2022, doi:10.3390/biomedicines11010088_

Round 1

Reviewer 1 Report

I have enjoyed reading the manuscript entitled: HATs and HDACs targeted protein acetylation in the occurrence and treatment of epilepsy by Wang. This paper primarily focuses on the changes of protein acetylation modification controlled by HATs and HDACs in the occurrence of epilepsy. Moreover, investigations of histone deacetylase inhibitors (HDACi) in the treatment of epilepsy are also summarized to provide new prospects for the clinical treatment of epilepsy. In general, the article is easy to read, quite well designed and can be of interest to readers and researchers. The methodologies are appropriate and aligned with the proposed objectives.

Further details and some suggestions on how to improve your work are described below:

-       Please check for double spacing.

-       Please summarize the main theme of the review in a graphical abstract.

-       Include images which makes the readers more interesting to read.

Author Response

Response: We appreciate the positive comments and suggestive advice from the reviewer. We have checked the double spacing. We included a graphical abstract to summarize the main theme of the review. In addition, we added a new figure (Figure 2) to make the manuscript more easily to be read.

Reviewer 2 Report

Your paper leaves me with a sense of vagueness in relation to how you think the acetylation – deacetylation processes you describe play a role in epileptogenesis. I may be incorrect, but I think you have provided very little clinical or experimental evidence that these changes occur before a first epileptic seizure has occurred, particularly in the animal models, and the report of such changes in a glioma associated with seizures may not support the possibility because the seizure generation will occur in surviving neurones in or near the tumour rather than the glioma tissue where I suspect the changes were found. Also, previous seizures may have induced the changes.

From your data I would have thought it more likely that the occurrence of seizures was responsible for the biochemical changes, though these may have then facilitated the occurrence of further seizures. If that were the case, was there evidence in the animal models of the time course of the development of the biochemical alterations? If the biochemical changes are secondary rather than primary they would still be of considerable importance and worthwhile treatment targets.

Basically, I think readers of your paper might be helped if you could provide, or at least discuss some extent, the way in which the biochemical changes might play a part in producing epileptogenesis, remembering that there are ‘many epilepsies’ and that the epileptogenic process may not be identical in all of them.

Author Response

Response: We appreciate the positive comments and suggestive advice from the reviewer. As far as we know, most research investigated the acetylation modification changes after seizure onset, while little clinical or experimental evidence of these changes before the first epileptic seizure has been reported in literatures. We added the discussion about the issue in the revised version. According to your suggestion, we deleted the content of acetylation modifications in glioma with seizures which was not an appropriate example in this situation.

Reviewer 3 Report

biomedicines-2047369: “HATs and HDACs targeted protein acetylation in the occurrence and treatment of epilepsy”

In this comprehensive review, the authors analyze a role of protein acetylation associated with activities of histone acetyltransferases (HATs) and histone deacetylases (HDACs) in mechanisms of epilepsy and in its treatment. The material is described successively, cogently and intelligibly with argumented conclusions. This manuscript could be recommended for publication after minor technical corrections.

Remarks/recommendations:

1)      a separate list of most important abbreviations would be very useful;

2)      the section 3.1 needs a few important references;

3)      line 283, “needs” seems to be correct;

4)      in the list of references: a) a dot between “Sclerosis” and “Cellular” should be recovered (line 455),

     b) in many references (66,80,86,96,97,104,114,121,122,140), the pages are omitted,

     c) ref. 146 should be rechecked.

5)      Figure 1 needs bigger fonts for “Ac”.

Author Response

Response: We appreciate the positive comments and suggestive advice from the reviewer. Thank you for pointing out the grammar mistakes and inaccurate cited references. We have revised the manuscript according to the suggestions and the detailed revisions are marked in the revised manuscript.

1) We listed the abbreviations in a new table (Table 3).

2) Some references were added in the Section 3.1.

3) We changed “need” to “needs”.

4) The references have been double checked.

5) The fonts for “Ac” were adjusted.